# Transcriptome Sequencing and Expression Analysis of Genes Related to Anthocyanin Biosynthesis in Leaves of *Malus* 'Profusion' Infected by Japanese Apple Rust

**Pengyuan Liu, Yilin Wang, Jiaxin Meng, Xian Zhang, Jing Zhou, Meiling Han, Chen Yang, Lingxin Gan and Houhua Li \***

Institute of Ornamental Plant, Northwest A & F University, Yangling 712100, China
**\*** Correspondence: lihouhua@nwafu.edu.cn; Tel.: +86-151-1480-0050

**Abstract:** Anthocyanins play many roles in plants, including providing protection from biotic and abiotic stresses. Japanese apple rust (*Gymnosporangium yamadae* Miyabe ex G. Yamada) causes serious diseases in plants of the genus *Malus* and results in reduced fruit production and quality. However, few studies have been done to unravel the molecular mechanisms of anthocyanin formation in rust-infected apple leaves. To identify new regulatory genes in apple leaves that may be involved in regulating rust-induced anthocyanin biosynthesis, we measured anthocyanin content and sequenced the transcriptomes of rust-infected and uninfected tissues of *Malus* 'Profusion' leaves. Significant color changes and anthocyanin enrichment (especially cyanidin-3-galactoside chloride) occurred in infected tissues, whereas no significant color change and a low anthocyanin level were observed in uninfected tissue. We identified 10,045 differentially expressed genes (DEGs) in these two tissue types, including 6021 genes that were upregulated in the infected tissue and 4024 genes that were downregulated. We also identified five structural genes that are putative regulators of anthocyanin biosynthesis. In addition, 56 *MYB* genes, 36 *bHLH* genes, and one *WD40* gene were identified among the obtained DEGs. According to the phylogeny of the amino acid sequences of transcription factors known to be involved in anthocyanin biosynthesis, one *MYB* gene (*MYB114-like*) and two *bHLH* genes (*bHLH33* and *bHLHA-like*) may relate to anthocyanin biosynthesis in rust-infected apple leaves. These data will provide insights into the molecular mechanisms underlying anthocyanin accumulation upon rust infection.

**Keywords:** transcriptome; *Malus*; anthocyanin; Japanese apple rust; differentially expressed genes; transcription factors

## 1. Introduction

*Malus* 'Profusion', an ornamental, deciduous crabapple tree or shrub in the Rosaceae family, is widely used in landscaping due to its beautiful colors and attractive shape. Like other members of the genus *Malus*, *Malus* 'Profusion' is vulnerable to Japanese apple rust [1]. The rust fungus, *Gymnosporangium yamadae* (Miyabe ex G. Yamada, 1904) infects apple trees and shrubs, causing serious diseases [2]. The *G. yamadae* fungal parasite afflicts coniferous trees, apple trees, and other species in the genus *Malus* [3]. In an earlier field experiment, we found obvious red patches on rust-infected tissues present on the leaves of *Malus* 'Profusion'. In addition, our preliminary physiological analysis revealed that the major red pigments in the red patches of rust-infected leaves of *Malus* 'Profusion' were anthocyanins. It is worth noting that Lu et al. also found that the content of flavonoids, especially anthocyanins, was significantly increased in the infected symptomatic tissue of apple leaves, and it was speculated that they play an important role in rust resistance in *Malus* [4].

The anthocyanin biosynthetic pathway has been well studied in plants. There are many enzymes that can catalyze anthocyanin synthesis, including chalcone synthetase (CHS), dihydroflavonol 4-reductase (DFR), and anthocyanin synthase (ANS) [5]. However, genes encoding these key enzymes (structural genes) are usually regulated by transcription factors, such as MYB, bHLH, and WD40, which are upstream regulators of anthocyanin metabolism [6]. These three transcription factors form the MYB-bHLH-WD40 transcription complex and regulate the biosynthesis of anthocyanin in plants [7]. Related studies indicate that MdMYB1, MdMYB10, and MdMYBA in the MYB transcription factor family play a regulatory role in the anthocyanin biosynthetic pathway in apple fruit (*Malus domestica* Borkh.) [8]. Studies using transgenic plants also showed that the expression of these genes plays a key role in anthocyanin accumulation and fruit coloration, and they cooperate with the MdbHLH3 and WD40 proteins to interact with promoters of the anthocyanin biosynthetic gene such as *MdANS* [9–11].

Anthocyanins are a water-soluble natural pigment belonging to the flavonoid family of organic molecules. They are also the main red pigments found in the Rosaceae family. Moreover, anthocyanins are associated with pathogen resistance. Laima et al. found that anthocyanin extracts from European *Viburnum* (cranberry) species had strong inhibitory effects on many common pathogenic bacteria [12]. Zhao et al. suggested that the antibacterial activity of purple corn extract is related to its anthocyanin content [13]. Another study showed that the differential regulation of the anthocyanin biosynthesis pathway resulted in the enrichment of anthocyanins in apple leaves, thereby improving resistance to scab and fire blight [14,15]. These results point to anthocyanins playing a role in defending against fungal pathogens in infected tissues.

Transcriptomic analysis is commonly used to identify differentially expressed genes (DEGs) and to quantify their relative expression in different tissues, including infected and uninfected leaf tissues [5]. In this study, by comparing the contents of anthocyanin-related metabolites and the expression levels of anthocyanin structural genes and associated transcription factors in rust-infected and uninfected leaf tissues of *Malus* 'Profusion' plants, we identified putative anthocyanin biosynthetic genes that participated in the response to rust infection in *Malus*. These results will guide the improvement of rust control and facilitate the selection of new rust-resistant *Malus* cultivars.

## 2. Materials and Methods

### 2.1. Plant Materials

Eight-year-old *Malus* 'Profusion' plants were used as the study material. The plants were grown in the crabapple germplasm nursery of Northwest A&F University, Yangling, China, maintained by standard horticultural management and infected with Japanese apple rust (*Gymnosporangium yamadae* Miyabe ex G. Yamada) under natural conditions. On 8 June 2018, three trees ($n$ = 3; considered as biological replicates) separated by a distance of three meters, were marked and from each tree, three visibly healthy and three rust-infected leaves were collected (total n of leaves per tree = 6; total nr of leaves harvested for analyses = 18). The healthy and infected leaves from each tree were respectively pooled to composite samples for the analyses ($n$ = 3). The harvested leaf samples were washed three times with distilled water. Red infected tissues (RIT) were collected from the red area of infected leaves by sterile blades, and uninfected leaf tissues (UIT) were collected from healthy leaves (Figure 1A). The samples were then immediately frozen in liquid nitrogen and stored at −80 °C for RNA extraction and the analyses of total anthocyanin and monomer phenol content.

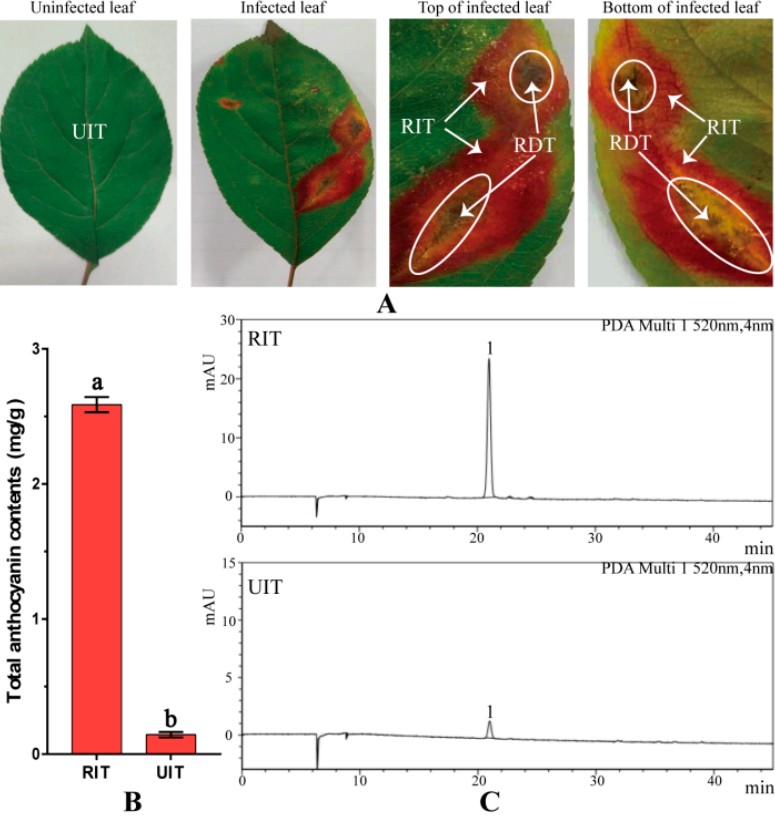

**Figure 1.** Images of *Malus* 'Profusion' leaf and anthocyanin contents in rust-infected red leaf tissue (RIT) and uninfected tissue (UIT). Shown are (**A**) Images of *M*. 'Profusion' leaves. Ellipses indicate rust-infected diseased leaf tissue (RDT); (**B**) Total anthocyanin content in RIT and UIT. (Data shown are the means ± SD of three biological replicates. The different letters represent significant differences between UIT and RIT, as judged by the paired *t*-test, $p < 0.05$); (**C**) HPLC and photo-diode array (PDA) trace images for anthocyanin in RIT and UIT (detected at 520 nm; peak 1 indicates cyanidin-3-galactoside chloride).

### 2.2. Total Anthocyanin Content

First, 0.1 g lyophilized leaf sample was weighed and soaked in 10 mL extraction solution (97:3 methanol: hydrochloric acid). The samples were then incubated at 4 °C for 48 h in the dark and extracted using ultrasound for 1 h. Next, we centrifuged the mixture at 6000 rpm for three minutes and collected the supernatant. Total anthocyanin content was determined by measuring absorbance on a UV-3802 spectrophotometer (Unico, Dayton, OH, USA) and applying the following formula: $A530 - 0.25 \times A657$. Cyanidin chloride (>95% HPLC, Sigma, St. Louis, MO, USA) was used to generate the calibration curves. Finally, total anthocyanin content was normalized to the dry weight of each sample.

### 2.3. Anthocyanin Biosynthesis-Related Compounds Content

First, we weighed out 0.4 g of the lyophilized leaves, soaked them in 8 mL of methanol at 4 °C for 48 h in the dark, and stirred them using ultrasonic waves for 1 h. Next, we centrifuged the mixture at 6000 rpm for three minutes and extracted the supernatant, which was then set aside. The substances within the samples were separated using a Shimadzu LC-20AT high-performance liquid chromatograph (HPLC) and a Lachrom-C18 column (4.6 mm × 250 mm, 5 μm, Hitachi, Tokyo, Japan). The proposed anthocyanin biosynthesis-related compounds peak was isolated by several analytical HPLC runs. The structure of the unknown substance was determined using LC–MS by comparing it with a standard as described by Li et al. [16]. Mobile phase A contained 0.04% aqueous formic acid and phase B contained chromatographic grade acetonitrile. The gradient elution was

as follows: 0–40 min, 95% A; 40–45 min, 60% A; 45–60 min, 0% A. The column temperature was maintained at 40 °C, the injection volume used was 10 μL, the flow velocity was 0.5 mL/min, and the detection wavelengths used were 280, 360, and 520 nm. All samples were filtered using a 0.22 μm filter before the analysis. Three technical replicates of each sample were analyzed. The type and concentration of the anthocyanin biosynthesis-related compounds in each extract were determined, and the content of each compound was calculated using the peak area normalization method.

### 2.4. RNA Extraction, Library Construction, and De Novo Transcriptome Assembly

Total RNA from both the RIT and UIT samples was extracted using the RNA prep polysaccharide polyphenol plant total RNA extraction kit (Tiangen Biotech, Beijing, China). The purity, density, and integrity of RNA samples was determined using a Nanodrop microvolume spectrophotometer, a Qubit 2.0 fluorometer, and an Agilent 2100 bioanalyzer to ensure that RNA samples were of sufficient quality for transcriptome sequencing. A library was assembled with high-quality RNA, and qualified libraries were sequenced using Sequencing by Synthesis (SBS) on an Illumina HiSeqX sequencing platform to obtain raw sequence data. Library construction and sequencing was performed by Beijing Baimax Technology Co., Ltd., Beijing, China.

The original sequencing data were cleaned by removing the adapter sequences and low-quality reads. The resulting clean reads were assembled into a transcriptome using the apple (*Malus domestica*) genome published by Nicolas et al. as the reference genome [17]. The reads were used for comparison and assembly using the HISAT2 [18] and StringTie [19] systems. The expression levels of each unigene were normalized to fragments per kilobase of transcript per million mapped reads (FPKM) [20]. The Pearson correlation coefficient *r* was used to evaluate correlations among replicates and among samples, and the rationality of sample selection was tested by examining the coefficient of determination ($r^2$).

### 2.5. Gene Identification, Gene Annotation, and Analysis of DEGs

Analysis of the differentially expressed genes was performed as per the DeSeq method of Wei et al. [21]. DEGs were identified using the following threshold criteria: corrected *p*-value (equivalent of false discovery rate; FDR) <0.05 (using Benjamini–Hochberg correction methods) and a fold change >2 or <−2. Functional annotation and classification was conducted by BLASTing (BLAST: Basic Local Alignment Search Tool) the gene sequences against seven public gene and protein databases including the nonredundant protein (Nr) database, the Swiss-Prot protein (SwissProt) database, the Gene Ontology (GO) database, the Eukaryotic Orthologous Groups of proteins (KOG) database, the Kyoto Encyclopedia of Genes and Genomes (KEGG) database, the Protein family (Pfam) database, and the Cluster of Orthologous Groups of proteins database (COG). GO hierarchical clustering was performed with the topGO R software package [22]. The enrichment of a specific GO term was considered significant when the D (distance) value ≤0.01 using the Kolmogorov–Smirnov test.

### 2.6. Phylogenetic Analysis and Sequence Alignment

The protein sequences of *MYB*, *bHLH*, and *WD40* were obtained from the NCBI (National Center for Biotechnology Information) non-redundant protein (Nr) database. Phylogenetic analysis was conducted using MEGA version 6 using the neighbor-joining method with 1000 bootstrap replicates. Sequence alignment was performed in DNAMAN Version 6.

### 2.7. Quantitative Real Time PCR (qRT-PCR) Analysis

Total RNA was extracted from the RIT and UIT samples, respectively. We used RNase-free DNaseI (Tiangen, Beijing, China) as per the manufacturer's protocol to remove genomic DNA. Next, we used a PrimeScript RT reagent kit (Takara, Otsu, Japan) to reverse-transcribe 1 μg of total RNA. Primers for qRT-PCR were designed and are listed in Table S1. 18S rRNA (DQ341382) was used as the internal control and UIT was used as the contrast group for normalization processing. In addition, the

relative expression level of each gene of UIT was set to 1. The relative expression level of DEGs in the samples was determined using 2× SYBR real-time PCR mixture kit (BioTeKe, Beijing, China) on the StepOnePlus real-time PCR system (Applied Biosystems, Waltham, MA, USA). The reaction procedure was a three-step process as follows: The first step was 94 °C for 2 min, the second step was 40 cycles of 94 °C for 15 s and 60 °C for 15 s, and the third step was 95 °C for 15 s, 60 °C for 1 min, and 95 °C for 15 s. The $2^{-\Delta\Delta Ct}$ method was used to calculate the relative expression levels of each gene [23].

### 2.8. Data Analysis

All experimental data were analyzed using paired *t*-tests, and $p < 0.05$ was defined as the threshold of statistical significance. Statistical tests were performed using IBM SPSS Statistics version 22 (IBM SPSS, Chicago, IL, USA), while graphical analysis and plotting were performed using GraphPad Prism Version 7 (GraphPad Software, San Diego, CA, USA).

## 3. Results

### 3.1. Analysis of Total Anthocyanin Content and Composition

The rust-infected red leaf tissues (RIT) were isolated from rust-infected diseased tissues (RDT) and uninfected leaf tissue (UIT) (Figure 1A). UIT was used as the control for all anthocyanin analyses. The total anthocyanin content of RIT was 18.50 times greater than that in UIT (Figure 1B). Moreover, HPLC data revealed that the cyanidin-3-galactoside chloride content in RIT was 15.80-fold greater than that in UIT, which was similar to the magnitude of the difference in total anthocyanin between infected and uninfected tissues. The most abundant anthocyanin in *Malus* 'Profusion' leaves was cyanidin-3-galactoside chloride (Figure 1C). In addition, naringenin, dihydroquercetin, catechin, and epicatechin, which are the main byproducts of the anthocyanin biosynthesis pathway, were more abundant in RIT than in UIT (by a factor of 1.57, 2.38, 2.59, and 4.89 times, respectively; Table 1). The levels of these metabolites were significantly different between the UIT and RIT, as revealed by the paired *t*-test ($p < 0.05$).

**Table 1.** Type and content of anthocyanin biosynthesis-related compounds in UIT and RIT samples.

| Compound | UIT/mg·kg$^{-1}$ | RIT/mg·kg$^{-1}$ |
|---|---|---|
| Catechin | 72.63 ± 1.00 b | 188.45 ± 0.45 a |
| Chlorogenic acid | 513.59 ± 12.56 b | 1947.46 ± 142.48 a |
| Coumaric acid | 107.24 ± 2.94 b | 171.26 ± 4.82 a |
| Cyanidin-3-galactoside chloride | 44.71 ± 0.48 b | 706.59 ± 7.52 a |
| Dihydromyricetin | 281.39 ± 7.62 b | 384.08 ± 16.89 a |
| Dihydroquercetin | 1485.13 ± 72.57 b | 3525.75 ± 61.70 a |
| Epicatechin | 54.89 ± 2.45 b | 268.37 ± 0.41 a |
| Eriodictyol | 271.32 ± 7.08 a | 278.33 ± 8.13 a |
| Naringenin | 1222.09 ± 31.85 b | 1915.27 ± 100.36 a |
| Phloretin | 10,404.03 ± 116.34 b | 28,553.27 ± 576.97 a |
| Proanthocyanin B1 | 92.67 ± 2.47 b | 123.94 ± 3.19 a |
| Proanthocyanin B2 | 4.56 ± 0.15 b | 38.50 ± 0.53 a |

Data are the mean values ± standard deviation (SD) of three biological replicates. The different letters represent significant differences between UIT and RIT, as judged by the paired *t*-test ($p < 0.05$).

### 3.2. Library Construction and Transcriptome Sequencing

We constructed six UIT and RIT transcriptome libraries based on three biological replicates for each tissue type. We obtained 49.14 Gb high-quality data after removing adapter sequences and low-quality reads. Bases with a Q score of 30 or higher (i.e., base call accuracy ≥ 99.9%) accounted for more than 93% of all the bases. The GC content (the ratio of Guanine and Cytosine content to the total nucleobase) ranged from 46%–50%, and the average contrast ratio was 70% (Table 2). These results confirmed the suitability of the obtained transcriptomic data for further analyses.

Pearson correlation coefficients (Pearson's *r* values) were used to determine the correlation in gene expression level (log10 (FPKM+1)) between tissue samples and among biological replicates for each tissue. The closer the $r^2$ values were to 1, the stronger the correlation between the two samples. As shown in Figure 2, the $r^2$ values between samples of the RIT and UIT tissues were less than 0.4, while the correlations between different biological replicates of the same tissue were over 0.99. These results suggested that the samples have good repeatability.

**Table 2.** Descriptive statistics of sequencing data.

| Sample Name | Clean Reads | Clean Bases | Q30 | GC Content |
|---|---|---|---|---|
| RIT1 | 28,534,074 | 8,533,976,408 | 94.10% | 50.04% |
| RIT2 | 22,858,851 | 6,835,393,636 | 93.71% | 50.01% |
| RIT3 | 27,789,895 | 8,305,725,370 | 93.40% | 49.84% |
| UIT1 | 32,429,472 | 9,701,511,274 | 93.58% | 46.84% |
| UIT2 | 27,360,624 | 8,189,892,014 | 93.46% | 47.18% |
| UIT3 | 25,312,509 | 7,573,777,970 | 93.97% | 46.79% |

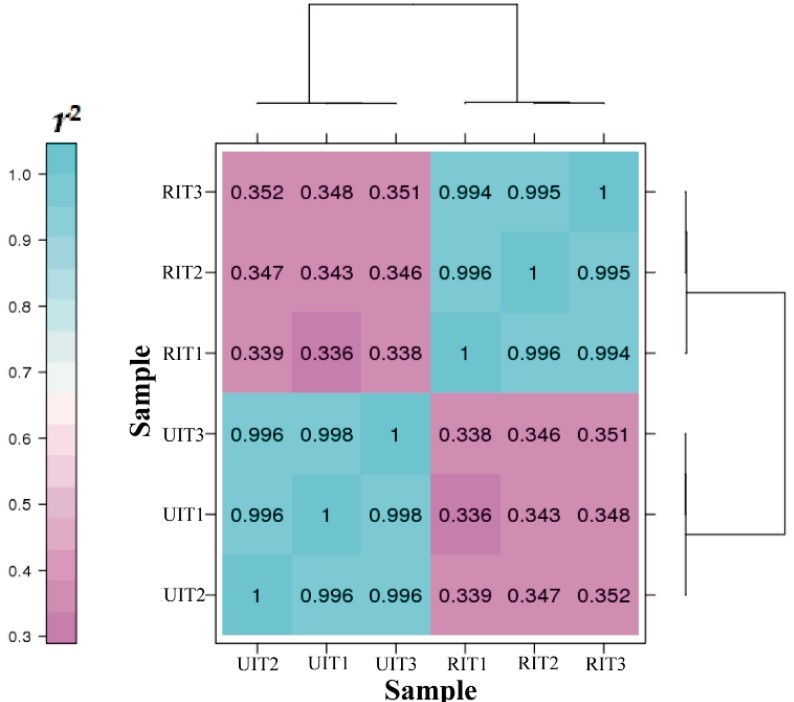

**Figure 2.** Correlation heat map of the gene expression quantity among biological replicates of the UIT and RIT samples (*n* = 3).

### 3.3. Annotation and Identification of DEGs

We considered genes that differed in expression between UIT and RIT samples at corrected *p* < 0.05 (using Benjamini–Hochberg correction methods) and had a minimum difference in expression of >(2)-fold or <(−2)-fold to be differentially expressed genes (DEGs). To show the overall distribution of the DEGs, we performed hierarchical cluster analysis and constructed volcano and minus average (MA) maps (Figure 3). After screening, we identified 10,045 DEGs. These included 6021 genes that were upregulated in RIT relative to UIT and 4024 genes that were downregulated. Next, a total of 9479 unigenes were annotated based on BLAST searches against seven public databases (Table 3). Only 296 of the 10,045 DEGs were not annotated, and therefore the total annotation rate was 97.05%.

**Table 3.** Annotation of non-redundant unigenes using queries of public databases.

| DEG Set | Total [1] | COG | GO | KEGG | KOG | NR | Pfam | Swiss-Prot |
|---------|-----------|-----|-----|------|-----|-----|------|-----------|
| RIT vs. UIT Genes [2] | 9749 | 4242 | 3950 | 3609 | 4978 | 9628 | 8204 | 7818 |
| Annotated (%) | 100.0 | 43.5 | 40.5 | 37.0 | 51.1 | 98.8 | 84.2 | 80.2 |

[1] Total represents the total number of annotated unigenes. [2] The percentages of annotated genes indicate that different types of data account for the total number of annotated unigenes.

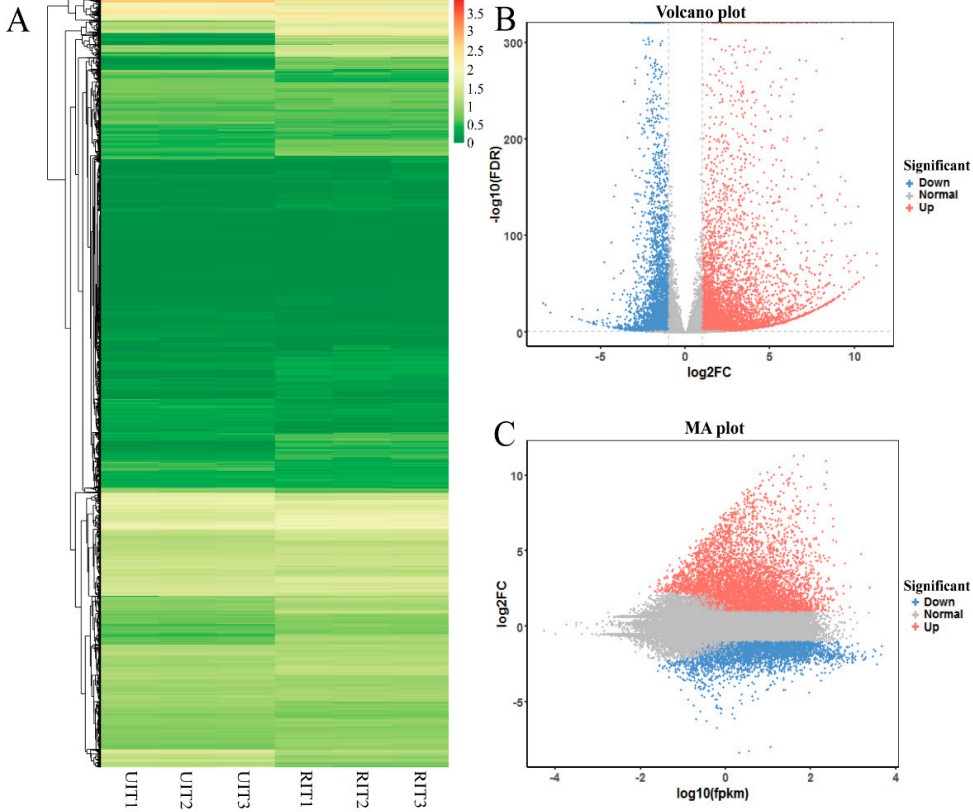

**Figure 3.** Expression analysis of the differentially expressed genes (DEGs) that were upregulated in RIT relative to UIT are shown in red, and downregulated genes are shown in green. (**A**) Cluster map of the DEGs in different biological replicates of RIT and UIT. Scale shows the log10 (FPKM+1) gene expression level in the sample; (**B**) DEG volcano plot; (**C**) DEG minus average (MA) plot.

### 3.4. Analysis of DEGs

### 3.4.1. GO Enrichment Analysis of the DEGs

We performed a GO enrichment analysis of all DEGs. GO classifications are divided into three major categories: biological processes, cellular components, and molecular functions. These categories are subdivided further. In our research, 8374 DEGs were assigned to the 'biological processes' category; 4341 DEGs were assigned to the 'cellular components' category; and 6141 DEGs were assigned to the 'molecular functions' category. These DEGs were further classified into 52 functional subcategories (Figure 4). Next, in order to identify significantly enriched GO terms for DEGs, we conducted a hierarchical clustering analysis on the DEGs and listed the ten most significantly enriched GO terms in Table 4. In total, 142 GO terms were significantly enriched with a D value ≤0.01 (Table S3). Photosynthesis (GO_ID: 0015979), oxidation-reduction processes (GO_ID: 0055114), and photosynthesis and light harvesting (GO_ID: 0009765) were the three most significant GO enrichment terms associated with the DEGs (Table 4). Each of these processes may be involved in anthocyanin biosynthesis.

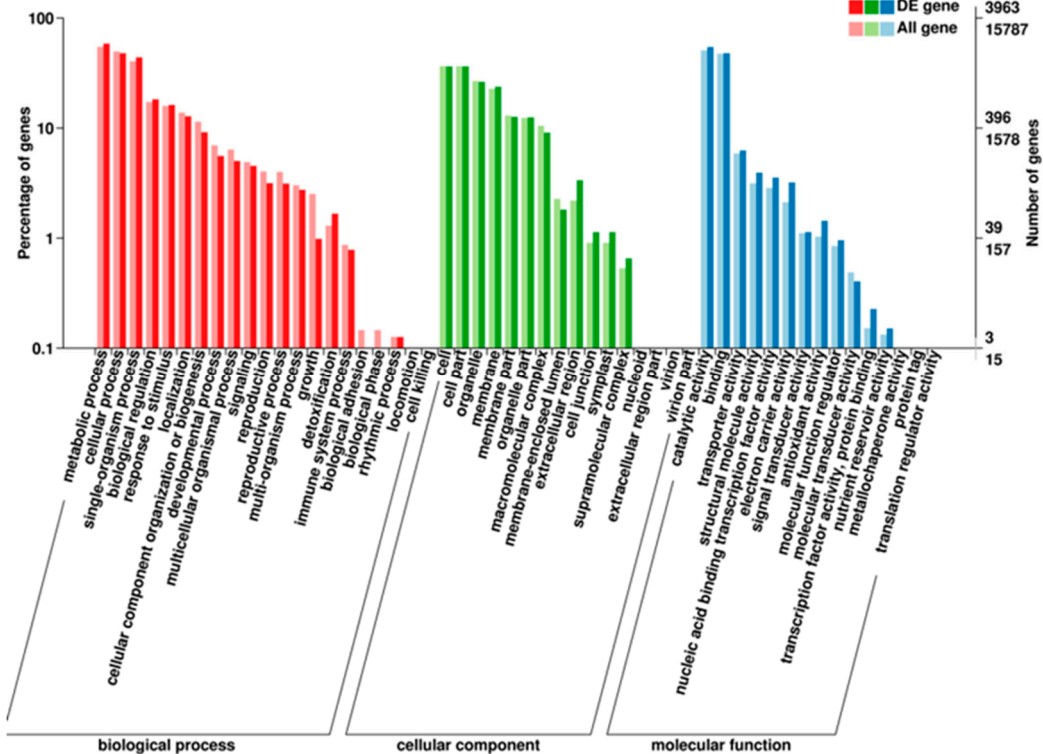

**Figure 4.** Gene Ontology (GO) classifications of the DEGs.

**Table 4.** The most prevalent GO enrichment results among the differentially expressed genes.

| GO_ID | Term | Annotated | Significant [1] | D Value [2] |
|---|---|---|---|---|
| GO: 0015979 | photosynthesis | 242 | 152 | $3.60 \times 10^{-20}$ |
| GO: 0055114 | oxidation-reduction process | 1726 | 603 | $6.90 \times 10^{-13}$ |
| GO: 0009765 | photosynthesis, light harvesting | 35 | 26 | $7.50 \times 10^{-11}$ |
| GO: 0006098 | pentose phosphate shunt | 103 | 52 | $2.60 \times 10^{-10}$ |
| GO: 0010027 | thylakoid membrane organization | 97 | 51 | $6.40 \times 10^{-10}$ |
| GO: 0019252 | starch biosynthetic process | 77 | 41 | $1.90 \times 10^{-8}$ |
| GO: 0000023 | maltose metabolic process | 62 | 35 | $1.90 \times 10^{-8}$ |
| GO: 0010207 | photosystem II assembly | 65 | 34 | $2.10 \times 10^{-7}$ |
| GO: 0035304 | regulation of protein dephosphorylation | 35 | 23 | $7.00 \times 10^{-7}$ |

[1] The number of significant DEGs under each GO term; [2] D value represents the conspicuousness statistic of the enriched term. The smaller the D value, the more significant the enrichment.

### 3.4.2. KEGG Enrichment Analysis of the DEGs

We mapped the identified DEGs to 124 pathways in the KEGG database. The 20 pathways that showed the most significant enrichment are shown in Figure 5. Of these 20 terms, seven were related to metabolism, seven were related to biosynthesis, three were related to photosynthesis, and the remaining three were related to the glycolysis/gluconeogenesis, ribosome, and pentose phosphate pathway, respectively. Of the 20 most significantly enriched pathways, the three most enriched pathways were photosynthesis-antenna proteins, flavonoid biosynthesis, and photosynthesis. The flavonoid synthesis pathway includes many structural genes involved in anthocyanin biosynthesis. We then used KEGG annotation to analyze the function of these DEGs and identified seven structural genes that may participate in anthocyanin biosynthesis in response to rust infection (Table 5). Of these, MD04G1003300 and MD13G1285100 were identified as chalcone synthases (CHS); MD06G1071600 and MD03G1001100 were identified as leucoanthocyanidin dioxygenases (LDOX/ANS); and MD01G1118100, MD15G1353800, and MD15G1024100 were identified as chalcone isomerase (CHI), flavanone-3-hydroxylase (F3H), and dihydroflavonol 4-reductase (DFR), respectively. This gene set encodes most of the key enzymes involved in the anthocyanin biosynthesis pathway [24].

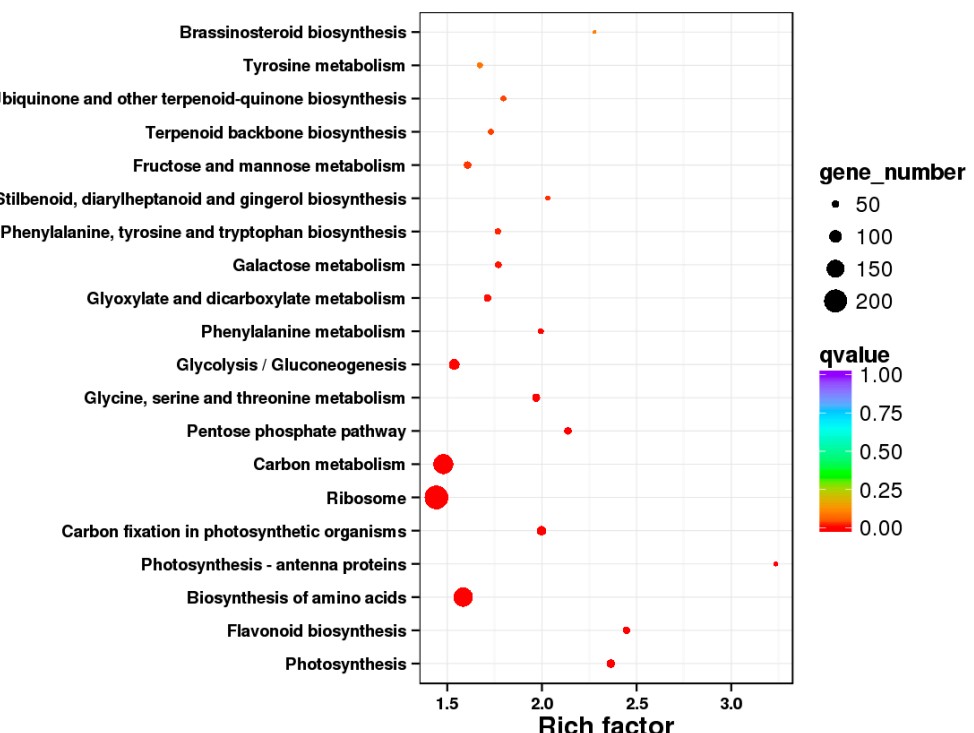

**Figure 5.** Bubble plot of the Kyoto Encyclopedia of Genes and Genomes (KEGG) database pathway enrichment of the DEGs. The first 20 enriched pathways are shown.

**Table 5.** Putative structural genes involved in anthocyanin biosynthesis identified from transcriptome data.

| GeneID | log2 Ratio (RIT/UIT) | FDR | Annotation |
|---|---|---|---|
| MD13G1285100 | 3.10 | 0 | chalcone synthase [*Malus domestica* Borkh.] |
| MD04G1003300 | 4.16 | 0 | chalcone synthase [*Sorbus aucuparia* Linn.] |
| MD01G1118100 | 2.10 | $2.91 \times 10^{-42}$ | chalcone isomerase [*Pyrus pyrifolia* (Burm. f.) Nakai.] |
| MD15G1353800 | 2.52 | 0 | flavanone 3-hydroxylase [*Prunus avium* Linn.] |
| MD15G1024100 | 3.00 | 0 | dihydroflavonol 4-reductase [*Malus domestica* Borkh.] |
| MD03G1001100 | 3.12 | 0 | leucoanthocyanidin dioxygenase [*Malus domestica* Borkh.] |
| MD06G1071600 | 2.69 | 0 | leucoanthocyanidin dioxygenase [*Malus domestica* Borkh.] |

### 3.4.3. Analysis of Transcription Factors in Identified DEGs

Anthocyanin synthesis is known to involve many transcription factors (TFs), including those of the MYB, bHLH, and WD40 families. Some members of these families can form a ternary complex (MBW) that can regulate the expression of anthocyanin synthase [7]. In this study, we found 56 *MYB* genes, 36 *bHLH* genes, and one *WD40* gene among the identified DEGs (Table S2).

To screen for *MYB* genes associated with anthocyanin biosynthesis, we conducted a phylogenetic analysis of the deduced amino acid sequences of the 56 *MYB* genes from the DEGs and other *MYB* genes that are known to participate in anthocyanin biosynthesis in plants. This analysis was performed according to the protocol published by Luo et al. [5]. One gene, *MYB114-like* (MD17G1261000),

showed a relatively higher degree of homology with other *MYB*s involved in anthocyanin biosynthesis, including *MCMYB10*, *MrMYB1*, and *NtAN2* (Figure 6), which have been proven to promote anthocyanin biosynthesis [25–27]. The amino acid sequence of MD17G1261000 was analyzed and compared to other *MYB* genes associated with anthocyanin biosynthesis. A highly conserved R2R3 sequence was found at the N-terminus of MD17G1261000 (Figure 7), which indicated that MD17G1261000 is a R2R3MYB transcription factor. To date, studies in diverse species have agreed that anthocyanin biosynthesis is regulated by a complex that includes R2R3MYB [28]. Meanwhile, another highly conserved $[D/E]L \times _2[R/K] \times _3L \times _6L \times _3R$ sequence has been found in the R3 domain of MD17G1261000 (Figure 7). This conserved sequence was found to bind to an R-like bHLH protein [29]. In addition, a conserved "ANDV" sequence was also found in the R3 domain of MD17G1261000 (Figure 7). This is a signature MYB-associated motif known to promote anthocyanin biosynthesis [30]. Finally, we found a short conserved motif (motif 6) in the C-terminal of MD17G1261000 (Figure 7), which was previously identified as a marker of positive regulation of anthocyanin biosynthesis in vegetative cells and/or reproductive tissues [30]. Taken together, these data indicate that MD17G1261000 may be a R2R3MYB transcription factor that recruits the R-like bHLH proteins, thereby activating anthocyanin biosynthesis.

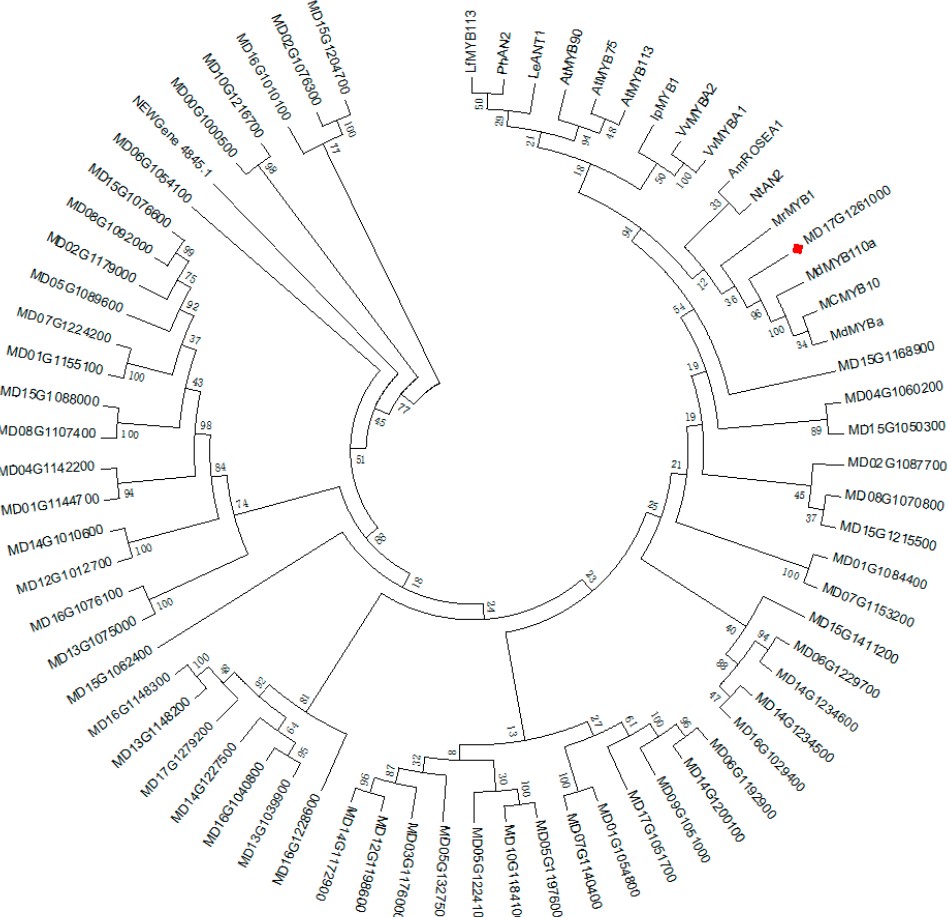

**Figure 6.** Phylogenetic analysis of the *MYB* genes involved in the regulation of anthocyanin biosynthesis in *M.* 'Profusion' and other plants. GenBank accession numbers for the *MYB* genes are as follows: *LfMYB113* (ACO52472.1), *PhAN2* (BAP28593.1), *LeANT1* (AAQ55181.1), *AtMYB90* (NM105310), *AtMYB75* (NP176057.1), *AtMYB113* (NP176811.1), *IpMYB1* (ABW69685.1), *VvMYBA2* (ABL14065.1), *VvMYBA1* (BAD18977.1), *NtAN2* (AQM49950.1), *MrMYB1* (ADG21957.1), *MdMYB110a* (NP001315777), *MdMYBA* (BAF80582), *MCMYB10* (JX162681), and *AmROSEA1* (ABB83826.1). Latin comparison table of the known *MYB* genes that regulate anthocyanin biosynthesis is shown in Table S4.

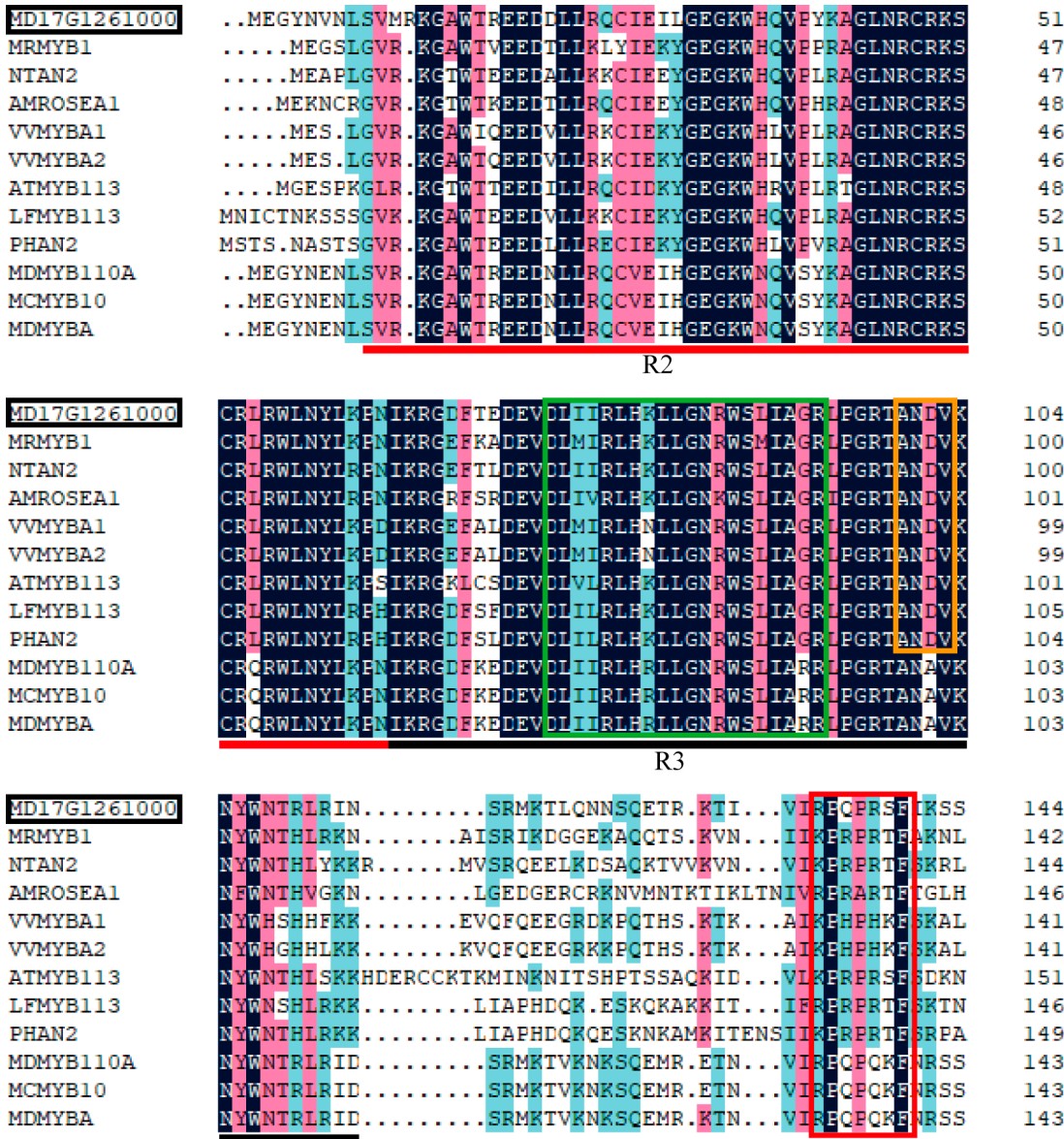

**Figure 7.** Protein sequence alignment of MD17G1261000 with *MYB* genes that promote anthocyanin biosynthesis in various plant species. The red line indicates the R2 domain and the black line indicates the R3 domain. The green, orange, and red boxes represent the [D/E]L × $_2$[R/K] × $_3$L × $_6$L × $_3$R sequence, the 'ANDV' sequence, and motif 6, respectively.

Next, we performed a similar phylogenetic analysis on the identified 36 *bHLH* genes (from the DEGs) and other *bHLH*s genes that are known to be involved in anthocyanin biosynthesis in plants. Two DEGs, *bHLHA-like* (MD03G126300) and *bHLH33* (MD07G1137500), from this study were found to be in the same clade as the *bHLH* genes known to regulate anthocyanin biosynthesis (Figure 8). Genes clustered into the same clade as MD03G126300, including *MdbHLH3*, *MdbHLH33*, and *NtAn1a*, were proven to be *bHLH* genes involved in anthocyanin biosynthesis [31,32]. The deduced amino acid sequence alignment showed that MD03G126300 and MD07G1137500 contained box 11, 18, and 13 at the N-terminal domain (Figure 9), all of which are involved in the interaction between *MYB* and *bHLH* [33]. These data show that MD03G126300 and MD07G1137500 probably interact with the *MYB*s. Among all DEGs, we identified only one *WD40* gene (*MSI4-like*, MD09G1193500) that may participate in anthocyanin biosynthesis; it was upregulated in RIT compared with UIT. However, the specific molecular mechanism needs to be further verified.

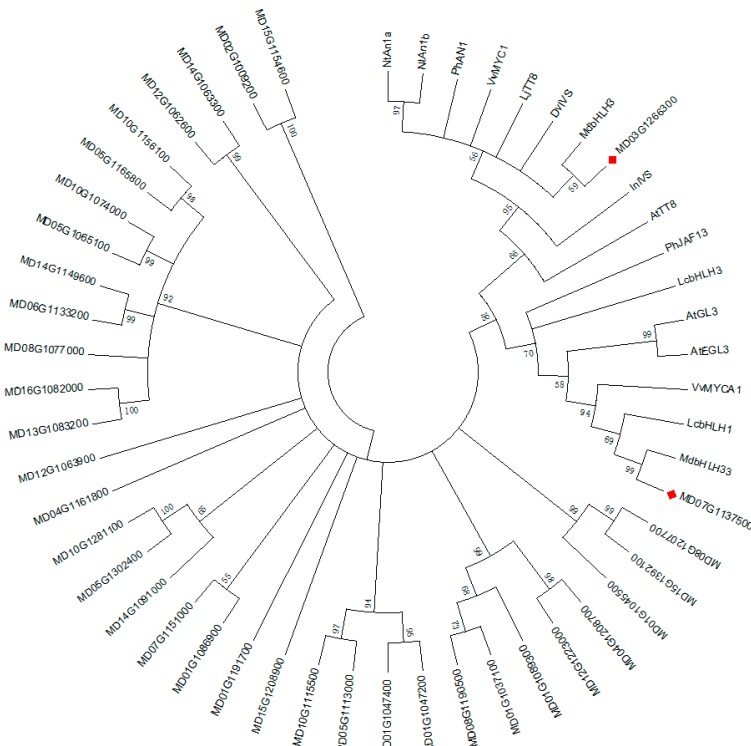

**Figure 8.** Phylogenetic analysis of the *bHLH* genes in *M*. 'Profusion' and the *bHLH* genes known to regulate anthocyanin biosynthesis in other plant species. GenBank accession numbers are as follows: *NtAn1a* (AEE99257), *NtAn1b* (AEE99258), *PhAN1* (AAG25927), *VvMYC1* (ACC68685), *LjTT8* (BAH28881), *InIVS* (BAE94394), *DvIVS* (AB601005), *AtTT8* (CAC14865), *AtGL3* (NP680372), *AtEGL3* (NP176552), *PhJAF13* (AAC39455), *VvMYCA1* (ABM92332), *LcbHLH3* (APP94124.1), *MdbHLH3* (ADL36597.1), *MdbHLH33* (ABB84474), and *LcbHLH1* (APP94122.1). Latin comparison table of the known *bHLH* genes that regulate anthocyanin biosynthesis is shown in Table S5.

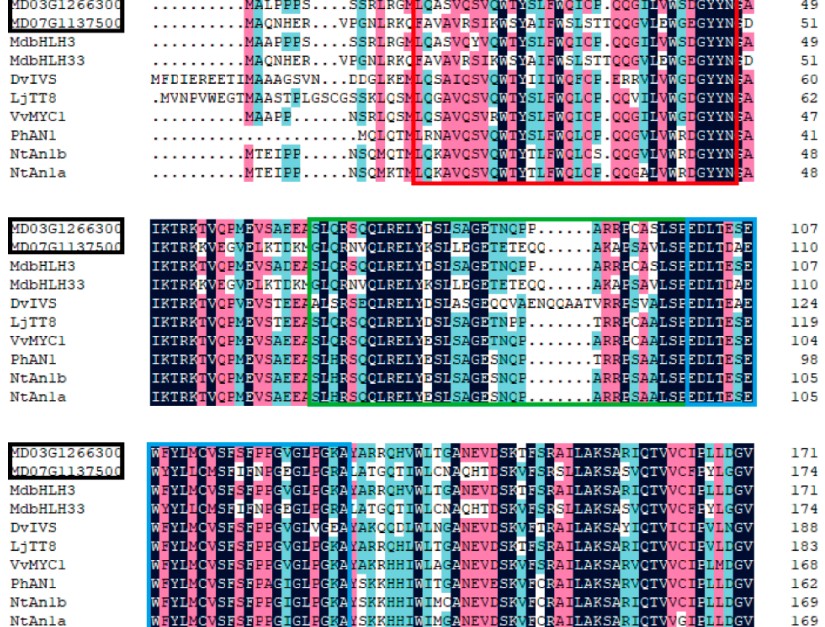

**Figure 9.** Alignment of the protein sequences of MD03G126300 and MD07G1137500 with *bHLH* genes known to promote anthocyanin biosynthesis in other plant species. The red, green, and blue boxes represent boxes 11, 18, and 13, respectively.

### 3.5. Quantitative Real-Time PCR (qRT-PCR)

To confirm the reliability of the transcriptome data, we examined the expression of eleven DEGs involved in anthocyanin biosynthesis in *Malus* 'Profusion' in the control and rust fungus-infected leaf samples. These DEGs included *CHS* (MD04G1003300), *CHI* (MD01G1118100), *F3H* (MD15G1353800), *DFR* (MD15G1024100), *ANS* (MD03G1001100), *MYB114-like* (MD17G1261000), *bHLHA-like* (MD03G126300), *bHLH33* (MD07G1137500), and *MSI4-like* (MD09G1193500). RT-PCR data indicated that changes in the expression of 11 genes in UIT and RIT were consistent with our transcriptome sequencing results. In addition, *MCMYB10*, which plays an important role in the normal development of leaf pigments in *Malus* spp. [34], was also verified with RT-PCR (Figure 10).

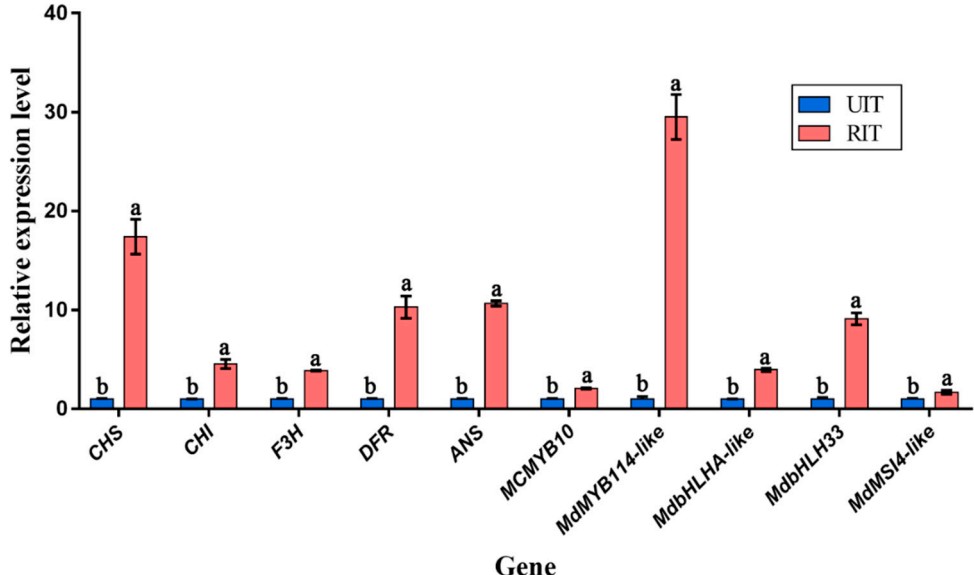

**Figure 10.** RT-PCR analysis of 11 DEGs related to anthocyanin biosynthesis in *M.* 'Profusion'. The data show mean values ± SD of three biological replicates. The different letters represent significant differences between UIT and RIT, as judged by the paired *t*-test ($p < 0.05$). 18S rRNA (DQ341382) was used as an internal reference gene for qRT-PCR data normalization, and the relative expression level of each gene in UIT was set to 1.

## 4. Discussion

Plants are affected by many environmental factors during their life cycle and have evolved many adaptive mechanisms to cope with adverse environments and external stress. When exposed to pathogens, plants initiate active defense responses that involve adjusting their physiology and producing bacteriostatic and antifungal metabolites [35,36]. These metabolites are often enriched in areas near the site of pathogen infection, resulting in differences in coloration between normal and infected tissues [37–39]. In this study, red lesions were observed around the rust-infected tissues on the leaves of *Malus* 'Profusion'. Such infections can be present as early as June. Pigment analysis found that the main chromogenic substance present in these red lesions was cyanidin-3-galactoside chloride.

Anthocyanins are important products of flavonoid metabolism. As colorful pigments, they not only attract pollinators and seed dispersers but also protect plants against pathogen infections [40]. Previous studies have shown that anthocyanins are involved in plant resistance to various fungal pathogens including *Botrytis cinerea*, bacterial blight, and gray mold [41–43]. Our results show that the content of cyanidin-3-galactoside chloride in RIT was significantly higher than that in UIT (Figure 1C). Moreover, in RIT, the contents of several flavonoids were also significantly higher than those in UIT (Table 1). Taken together, the evidence accumulated here suggests that the anthocyanin content of rust-infected leaves of *Malus* 'Profusion' may be an active defense mechanism adopted by plants to resist rust infection.

Anthocyanin biosynthesis is controlled by multiple structural enzymes, including CHS, CHI, F3H, DFR, and ANS. [44]. In this study, we identified seven anthocyanin biosynthetic genes as DEGs between the RIT and UIT samples. Among the structural genes verified by qRT-PCR, *ANS* is a key gene involved in anthocyanin biosynthesis. *ANS* can catalyze the conversion of the colorless leucoanthocyanidins to red cyanidin [45] and contribute to the redness of plant tissues. In this study, we found that the expression of *ANS* in the RIT samples was approximately 10.65-fold higher than that in UIT. Our results show that the higher expression of *ANS* in RIT is related to the redness of infected spots.

Structural genes associated with anthocyanin biosynthesis are regulated by transcription factors including those of the *MYB*, *bHLH*, and *WD40* gene families [46]. Some of these transcription factors can form a MYB-bHLH-WD40 ternary complex and jointly regulate the synthesis of anthocyanins [7]. Of the TFs regulating the anthocyanin synthesis pathway, MYB TFs are the most well studied [5]. In this study, 56 *MYB* genes were found among the identified DEGs. Phylogenetic analysis of the *MYB* genes in *M.* 'Profusion' and other plants resulted in the identification of *MYB114-like* (MD17G1261000), a gene that had the highest homology to the *MYB* genes known to regulate anthocyanin biosynthesis in other plant species. Further comparative analysis revealed that the N-termini of MD17G1261000 had a highly conserved R2R3 domain. In addition, MD17G1261000 also contained a conserved $[D/E]L \times_2 [R/K] \times_3 L \times_6 L \times_3 R$ structure, which was previously shown to be associated with *bHLH* binding [29]. We also found two motifs, an "ANDV" sequence and motif 6, in the R3 domain and the C-terminal of MD17G1261000, respectively. To date, motif 6 has only been found in *MYB* genes that promote anthocyanin biosynthesis in dicotyledons [30]. Another significant motif in *MYB* genes responsible for promoting anthocyanin biosynthesis is "ANDV". Here, the ANDV motif was found at positions 100–103 of the R2R3 domain in MD17G1261000 [30]. Taken together, our data indicate that MD17G1261000 may be a *R2R3-MYB* gene that recruits the R-like bHLH proteins to activate anthocyanin biosynthesis in response to rust infection, although more studies are needed for validation. In addition, typical box 11, 18, and 13 motifs were found in the protein sequences of the two selected bHLH transcription factors: bHLHA-like (MD03G126300) and bHLH33 (MD07G1137500). These structures play an important role in the interaction with MYB transcription factors, leading to enhanced transcription of anthocyanin biosynthetic genes [33]. These findings suggest that the two *bHLH* genes in *Malus* may also participate in the regulation of anthocyanin biosynthesis, but additional studies are needed to confirm this. Only one *WD40* gene, *MSI4-like* (MD09G1193500), was identified among the DEGs, and it was upregulated in the RIT samples compared with the UIT samples. Its involvement in anthocyanin biosynthesis remains to be elucidated.

In addition, we found that the level of anthocyanins in *Malus* 'Profusion' leaves, which is induced during normal development at a high carbon–nitrogen ratio, is regulated by *MYB10* [34,46,47]. qRT-PCR analysis revealed that the expression of *MYB10* in RIT samples was 2.08 times greater than in the UIT samples. Although *MYB10* may not be the main regulator, it has a role in the formation of red lesions. In addition, the expression of *MYB114-like* (MD17G1261000) in the RIT samples was 29.51-fold greater than that in the UIT samples, consistent with that observed for *ANS* expression. This result suggests that MYB114-like (MD17G1261000) may be the key MYB TF that regulates anthocyanin biosynthesis in response to rust infection.

Based on our results, we propose the molecular mechanism responsible for anthocyanin biosynthesis at sites of rust infection on the leaves of *Malus* 'Profusion'. The expression of *MYB114-like* (MD17G1261000) was activated in rust-infected areas. The resulting protein can combine with bHLH33 (MD07G1137500), bHLHA-like (MD03G126300), and MSI4-like (MD09G1193500) to form the MBW structure. The MBW complex promotes the expression of *ANS* and other genes in the infected area. This in turn makes the infected area appear red due to anthocyanin biosynthesis. The specific mechanism should be further characterized in future studies.

## 5. Conclusions

In summary, rust infection results in increased anthocyanin contents in infected apple leaves, suggesting that it plays a role in restricting the spread of the rust infection of these pigments in *Malus* leaves. Our results show that *MYB114-like* may be a key MYB TF that regulates anthocyanin biosynthesis in response to rust infection. bHLH33 and bHLHA-like may be key bHLH TFs that regulate anthocyanin biosynthesis in response to rust infection. These findings provide insights into the molecular mechanisms underlying anthocyanin accumulation upon rust infection. Japanese apple rust (*Gymnosporangium yamadae* Miyabe) is an alternate parasitic fungus. Its telial stage occurs on junipers. Spherical or hemispherical galls are formed on the infected junipers and harm the junipers' growth and development. Furthermore, rusts are a widespread threat to forestry plants such as pine, poplar, willows, spruce, etc. Improved knowledge about these mechanisms may help to select tree lines with improved resistance to Pucciniales rusts, many of which cause severe problems in forests and tree plantations across the world.

**Supplementary Materials:** The following are available online at http://www.mdpi.com/1999-4907/10/8/665/s1, Table S1: Primer sequences used for real-time quantitative PCR; Table S2: Transcription factors identified among the differentially regulated genes; Table S3: The results of GO terms significant enrichment among DEGs (KS ≤ 0.01); Table S4: Latin comparison table of the known *MYB* genes that regulate anthocyanin biosynthesis; Table S5: Latin comparison table of the known *bHLH* genes that regulate anthocyanin biosynthesis.

**Author Contributions:** P.L. performed the main experiments and data analysis and wrote the paper; Y.W., J.M. performed some experiments. X.Z. and J.Z. contributed to writing the paper. M.H., L.G., and C.Y. participated in the collation of part of the data; H.L. designed the experiments; and all authors read and approved the final manuscript.

**Funding:** This research was funded by the National Natural Science Foundation of China (NSFC) (31570697).

**Conflicts of Interest:** The authors declare no conflict of interest.

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
