# Peer review of "Transcriptome Sequencing and Expression Analysis of Genes Related to Anthocyanin Biosynthesis in Leaves of Malus ‘Profusion’ Infected by Japanese Apple Rust"

_forests, doi:10.3390/f10080665_

Round 1

Reviewer 1 Report

The manuscript contains many results. It is understandable and may contribute to the elucidation of tree-defense against rust infection.

I have some minor issues that should be addressed:

Lane 14, lane 39, 40, lane 67: The name (Gymnosporangium yamadai) is not clear. Is it Gymnosporangium yamadae(Miyabe ex G. Yamada, 1904)?

Lane 36: Rosaceaeis the family (change genus to family)

Lane 43-45: Was there a previous study done by your group that was not published?

Introduction in general: In my opinion it would be helpful to include some further information about MYB, bHLH and WD40 and about biosynthesis and regulation that is known so far.

Lane 59: “we characterize the mechanisms of anthocyanin biosynthesis” sounds too promising and should be changed to “we have identified genes that are probably involved in anthocyanin biosynthesis” or something like that.

Lane 307: Change Real-Time Quantitative PCR to Reverse-Transcription Quantitative PCR (qRT-PCR)

Figure 2 appears twice. Check all Figure numbers and adjust.

Table 5: What stands FDR for?

Why wasn`t Lu et al., 2017 (Flavonoid Accumulation Plays an Important Role in the Rust Resistance of Malus Plant Leaves) cited? That study seems to be very similar.

Lane 251: “…the MYB genes and other amino acid sequences…”

Lane 288: “Like other genes, x and y are exposed at the N-terminus…”

For a more scientific presentation, DNA and amino acid sequences shouldn`t be mixed. It would be more appropriate to use words like “geneproduct”. Check the complete manuscript.

Author Response

Dear Reviewer 1

Thank you for your positive comments.Please see the attachment.

Reviewer 2 Report

This manuscript investigates the genes responsible for the red pigmentation around rust infection in Malus.  The authors found genes that were differentially expressed between infected red tissue and uninfected green tissue, and highlight differentially expressed genes that seem to be involved in anthocyanin biosynthesis.  They also suggest potential MYB and bHLH transcription factors in Malus that may regulate anthocyanin biosynthesis based on phylogenetic analysis comparing differentially expressed MYB and bHLH genes from their analysis with those genes known to regulate anthocyanin biosynthesis in other species.

Specific comments:

Abstract:

-p.1, line 15-16 Does ‘landscape quality’ have something to do with fruit?  The way the sentence is worded makes it seem like it does.

Introduction:

-p. 1, line 42-p.2, line 43 Has there been a study that shows that the red areas prevent expansion of the pathogen?

-p. 2, lines 43-45 Citation?

-p. 2, line 46 There are many anthocyanin pigments, not just one

-p. 2. Line 53 Period should go before the space

Methods:

-p. 2, line 66 There is an extraneous ‘leaves’

-p. 2, lines 67-70 Were whole leaves used, or only the red portion of the leaves in rust-infected leaves?

-p.2, line 69 Were these replicates biological replicates, i.e. each leaf sample was from a different tree?  Three biological replicates is a minimum for differential expression studies.  Were there paired samples, infected and uninfected, from each tree?

-p. 4, line 143 Were all samples paired infected and uninfected from the same tree?  Where were the paired t-tests used?

Results:

-p. 4 lines 153-154, 159 and Table 1 Were statistics run on these to determine which compounds were statistically different in infected and uninfected tissue?

-p. 4, lines 177-178 Unclear what ‘gene expressive quantity values’ are.

-Fig. 2 Could the sample names be mislabeled?  The correlation between each sample with itself should be 1, but the diagonal line of 1s is on the wrong diagonal for the sample labelling.  E.g. the correlation for UIT1 and UIT1 in the upper left hand corner should be 1, not 0.352 and the correlation for RIT3 and UIT1 in the lower left hand corner should not be 1.

-p. 6, line 195 BLAST should not be used as a verb

-p. 6, line 196 don’t need comma after ‘(GO)’

-p. 7, line 202 Fig. 3 is labelled as a second Fig. 2.  Labelling and key for Fig. 3B&C is far too small.  Also red/green is not the best combination for colorblind individuals.

-p. 7, line 206 Table 3 Does this table represent only the DEGs that were annotated?  On p. 6, line 199, you state that 296 DEGs were not annotated.  Would it make sense to base the values in the table on the total DEGs to not mislead the reader?

-Figures are misnumbered starting from Fig. 3 (cluster map, volcano plot, MA plot) both in the text and in the figure legends

-p. 7, lines 209-210 Fig. 3 (GO enrichment histogram) Based on the histograms in Fig. 3, it doesn’t look like the DE genes are enriched in these categories compared to the all gene data. What metrics were used to determine significant enrichment?  These values should be reported.

-p. 7, line 214 It is unclear what the purpose of the hierarchical clustering analysis is.  Also, this analysis was not described in the methods.

-p. 9 line 247 ‘trinary’ should be ‘ternary’

-p. 9, lines 251-252 I would clarify that the 56 MYB genes are those identified as differentially expressed in this study along with MYB genes from other plants known to be involved in regulating anthocyanin biosynthesis

-p. 10 Fig. 5 (MYB tree) You indicated in the methods that you performed a bootstrap analysis.  You should report bootstrap values for each branch on the tree so the reader can evaluate how robust the relationships are.

-p. 10, lines 273-277 I would like to see a table, perhaps in supplemental, that lists which species all the known MYBs that regulate anthocyanin biosynthesis come from.

-p. 11, lines 283-285 I would explicitly state that you included the DEG bHLH genes from your study and bHLH genes known to be involved in anthocyanin biosynthesis from other plants.  Also, the correct phylogenetic terminology is that the two DEGs from this study are found in the same clade as the bHLH genes known to regulate anthocyanin biosynthesis.

-p. 11, line 286 Should be ‘genes located in the same clade as…’ not ‘same branch as…’. Each gene is on its own branch.  A clade refers to branches that share a common ancestor and all of its decendants.

-p. 11, line 291 Should be ‘data show’ not ‘data shows’

-p. 11, line 292 ‘interact’ may be more appropriate than ‘combine’

-p. 11, lines 292-295 You could potentially run a similar phylogenetic analysis to those run with MYB and bHLH genes if you include Malus WD40 genes from your sequencing analysis, even if those genes were not differentially expressed.  With the gene sequences, you could determine whether the DE WD40 gene you found was most closely related to WD40 gees known to be involved in anthocyanin biosynthesis in other species.  Just be sure to make it clear that only the one gene was differentially expressed in your analysis.

-p. 11, lines 293-295 The first part of this sentence is a fragment and should be revised.

-Fig. 7 Again, bootstrap values should be reported for every branch, and I would like to see a supplemental table with the species names for each gene.

-Fig. 9 Red/green is not the best for colorblind individuals.  Also, was the UIT value for each gene set to a relative expression level of 1?  It could be interesting to see the relative expression among these genes.  Either way, this should be stated in the figure legend and methods.

-p. 14, line 329 Should be ‘Anthocyanins are important products…’

-p. 14, line 330 Should be ‘protect’

-p. 14, line 343 I would say ‘involved in’ or something similar because ‘regulating’ has the connotation of transcription factors (like MYB, bHLH, and WD40 in this case).

-p. 15, line 354-355 I would say something like ‘to MYB genes shown to regulate anthocyanin biosynthesis in other species’

-p.15, line 359 and 360 Add ‘genes’ after ‘MYB’ (2x)

-p. 15, line 364 To be clear, perhaps add something like although further studies are need to confirm this

-p. 15, line 367 Should be ‘MYB’ not ‘MYBs’

-p. 15, line 368 Again, indicate that this evidence suggests that these genes in Malus may also be involved in regulation of anthocyanin biosynthesis , but additional studies are needed to confirm.

-p. 15, line 371-372 This is the first time that it is mentioned that MYB10 is involved in leaf reddening.  When discussed in the results, it was presented as normal pigment production in leaves.  I interpreted this as chlorophyll production, not anthocyanin production, so I would revise the section of the results that discusses MYB 10 and explicitly state hat it is involved in redness/anthocyanin production.

Author Response

Dear Reviewer 1:

Thank you for your comments.Please see the attachment.
